# The Potential of Simulating Energy Systems: The Multi Energy Systems Simulator Model

**Luigi Bottecchia** [1,2,*,†] 🆔**, Pietro Lubello** [3,*,†] 🆔**, Pietro Zambelli** [1] 🆔**, Carlo Carcasci** [3] 🆔 **and Lukas Kranzl** [2] 🆔

1   Institute of Renewable Energy, Eurac Research, Via A.Volta 13/A, 39100 Bolzano, Italy; pietro.zambelli@eurac.edu
2   Energy Economics Group, Institute of Energy Systems and Electrical Drives, Technische Universität Wien, Gusshausstraße 25-29/370-3, 1040 Vienna, Austria; lukas.kranzl@tuwien.ac.at
3   Department of Industrial Engineering, University of Florence, Via Santa Marta 3, 50139 Firenze, Italy; carlo.carcasci@unifi.it
*   Correspondence: luigi.bottecchia@eurac.edu (L.B.); pietro.lubello@unifi.it (P.L.)
†   These authors contributed equally to this work.

**Abstract:** Energy system modelling is an essential practice to assist a set of heterogeneous stakeholders in the process of defining an effective and efficient energy transition. From the analysis of a set of open-source energy system models, it emerged that most models employ an approach directed at finding the optimal solution for a given set of constraints. On the contrary, a simulation model is a representation of a system used to reproduce and understand its behaviour under given conditions without seeking an optimal solution. In this paper, a new open-source energy system model is presented. Multi Energy Systems Simulator (MESS) is a modular, multi-energy carrier, multi-node model that allows the investigation of non optimal solutions by simulating an energy system. The model was built for urban level analyses. However, each node can represent larger regions allowing wider spatial scales to be represented as well. In this work, the tool's features are presented through a comparison between MESS and Calliope, a state of the art optimization model, to analyse and highlight the differences between the two approaches, the potentialities of a simulation tool and possible areas for further development. The two models produced coherent results, showing differences that were tracked down to the different approaches. Based on the comparison conducted, general conclusions were drawn on the potential of simulating energy systems in terms of a more realistic description of smaller energy systems, lower computational times and increased opportunity for participatory processes in planning urban energy systems.

**Keywords:** energy system modelling; energy optimization; energy simulation; Multi Energy Systems Simulator (MESS)

## 1. Introduction

The decarbonization of the energy sector is one of the major challenges in confronting the climate emergency. To meet the Paris Agreement goals [1], the global energy system will have to undergo a profound transformation, shifting from being predominantly fossil fuel-based to relying on clean renewable energy sources, while guaranteeing sustainability, fairness and security of supply [2]. In this context, energy system modelling represents a valid support in the decision-making process of planning future energy systems. Numerous tools have been developed and employed in recent years and are continuously updated to face new emerging challenges and consider the development and upcoming of innovative technologies.

The Open Energy Modelling (openmod) [3] initiative provides a list of about 40 open-source energy system models characterized by different aims and structures. Most of the models listed are focused on the optimization of energy systems rather than on their simulation, performed without looking for an optimal solution. Several works can

be found in the literature reviewing challenges and trends of energy system modelling. Pfenninger et al. [4] exploited the challenges of energy system modelling.

They pointed out that it is essential to improve the transparency of the developed tools, which translates into adopting an open-source approach coupled with full access to the data used and a clear definition of system boundaries, variables and structures used in the model, to allow users and other experts to clearly understand how each model works. Moreover, the increasing relevance of renewable energy sources (RES) makes it necessary to use high temporal and spatial resolution data.

Another identified challenge regards the complexity of the models provided. In light of this, Pfenninger et al. suggested adopting approaches that look into non-optimal solutions as well. Indeed, using this approach should allow a more straightforward formulation of the mathematical problem.

Chang et al. [5] recently updated and expanded the work initiated by Pfenninger et al. and, by reviewing a set of modelling tools, identified current trends in the field of energy modelling. Specifically, they found that, in recent years, modellers have addressed some of the aforementioned challenges through increased cross-sectoral synergies, growing attention to open access and open-source publications as well as temporal resolution improvement. Moreover, among the identified trends, a limited amount of simulation tools have emerged compared to optimization ones.

### 1.1. Aim and Structure of the Paper

Considering the outlined context and the existing models, a novel open-source simulation model was developed with the intent to simulate multi-energy systems (i.e., considering multiple energy vectors and sectors), at an urban scale, without looking for an optimal solution. The tool was called the Multi Energy System Simulator (MESS). Given the necessity of simpler models that can improve transparency and clarity in energy system modelling and the limited amount of open-source simulation tools, the authors decided to investigate the potential of the simulation approach through MESS.

Moreover, we wanted to analyse how the differences in the approach between simulation and optimization reflect on different end-users. Hence, this work also focuses on understanding how simulation tools can lead to a more participatory and democratic process with greater involvement of both decision-makers and the community. At the same time, this paper presents the developed tool by explaining its structure and functioning aspects and comparing it to an existing optimization tool to depict the aforementioned aspects.

In summary, this paper aims at presenting a newly developed tool and at giving a contribution in answering to the following research questions:

1. What is the potential of simulating an energy system rather than optimizing it?
2. Does simulating an energy system lead to a more participatory and democratic process in planning future energy systems at the urban level?
3. How can MESS contribute to tackling the challenges of modelling the energy transition?

The structure of the paper is as follows. Section 2 provides the background information needed to clearly present the boundary conditions and the main definitions adopted. Section 3 introduces the main characteristics of the developed tool (MESS) and the details of how it works. Additionally, it introduces the rationale behind the comparison between MESS and an existing optimization tool (Calliope [6]). Section 4 illustrates the results of this comparison, while Section 5 builds on the results to discuss them in light of the research questions. Finally, Section 6 summarizes the main findings and presents areas of future development on the topic.

## 2. State of the Art and Definitions

Considering the current challenges of energy system modelling, one of the aims of this work is to address some of them within the development of a new simulation tool (MESS). Thus, to increase the transparency and the clarity of the work proposed, this

section presents the main background information. In this way, the readers can have a clear overview of the boundary conditions and the main definitions adopted.

The Fifth Assessment Report of the Intergovernmental Panel on Climate Change (IPCC) [7] defines an energy system as a system comprehending all components related to the production, conversion, delivery and use of energy. Similarly, Jaccard [8] refers to an energy system as the combined processes of acquiring and using energy in a given society or economy.

Building on these two definitions and following the terminology used by Pfenninger et al. [6], in this work, an energy system is considered as the combination of processes and technologies related to the production, consumption, conversion and transmission of energy in a given society, economy or location.

A location can be considered as a site that contains multiple technologies and also other locations. This means that a location can be either a single building with its own technologies, or as a district composed by multiple buildings. In this way, an energy system is not only seen from a mere technical point of view but also as including the spatial and socio-economic dimensions.

Given the definition of energy system, it is then necessary to define what a model is. In this sense, Rosen [9] considers a model as the formalized representation of a natural system with its own rules. Keirstead et al. [10] added that, within the energy and engineering fields, the formalization is to be intended in the form of mathematical models and computer codes. Hence, in this work, a model is considered as the representation of a system with its own rules through the use of a mathematical formulation.

In the context of energy system modelling, multiple approaches can be considered. One of the main distinctions is between optimization and simulation models. Building on the definition given by Wurbs [11], Lund et al. [12] consider an optimization approach as the one that makes use of a mathematical formulation to find the optimal solution of a given problem. The problem is generally defined by an objective function subject to multiple constraints.

Both the objective function and the constraints are dependant on a set of decision variables whose values are set during the optimization process. The objective function can be related to emissions, system costs or other aspects related to the system. On the contrary, both Wurbs and Lund et al. defined a simulation model as the representation of a system used to forecast its behaviour under certain given conditions. Both works highlighted that simulation models are meant to be used to understand the performance of a certain system under a given set of assumptions.

For the purpose of this work, we considered a simulation model as the representation of a system used to reproduce and understand its behaviour, under given conditions, without looking for an optimal solution. The slight difference is due to the authors' belief that the first purpose of simulation tools should be to reproduce the behaviour of a given system rather than to forecast it. Indeed, forecasting can be thought as a subsequent step, to be performed through scenario analysis or similar approaches. In this regard, a simulation model could be used to evaluate the consequences of a given choice whether it might be technical, political or social.

Provided that a common trend identified in the literature is the shift from single-building to urban scale analyses [13] and that one of the purposes of this work is to develop a tool that works at an urban scale, a clarification on the usage of the expression urban scale is given as follows. Eurostat provides common definitions for the European geographical areas starting from the concept of degree of urbanisation .

According to this definition, the degree of urbanisation provides a classification for local administrative units (LAUs) obtained from the combination of geographical proximity and population density [14]. The classification is made by considering a raster cell of 1 km$^2$. LAUs can then be: cities (densely populated areas), towns and suburbs (intermediate density areas) or rural areas (sparsely populated areas).

Urban areas are represented by the first two classes: cities and towns and suburbs [15]. At this point, the non-trivial aspect to consider is the integration of the *urban* concept in the energy system definition. Both Keirstead et al. and Alhamwi et al. [10,16] exploited the approach used by Ramaswami et al. [17], called *geographic-plus*, which considers not only the energy flows but also the geopolitical boundaries of a system.

Hence, in the current work, and more generally in the context of energy system modelling, an urban scale is considered the resolution incorporating districts and cities, while an urban area is an area with an intermediate or high density of population. In this way, a energy system model is considered to be able to perform analysis at the urban scale when it has a spatial resolution that goes down to the district level, allowing to consider urban areas composed by small, medium and large cities.

*Models Review*

As highlighted by Pfenninger et al. [4], a possible solution to overcome the issue of increasing complexity in energy system modelling is to be able to investigate non-optimal solutions. As for the definition given in Section 2, using a simulation approach allows exploring these kinds of solutions. Further developing the considerations made by Lund et al. [12], the authors decided to analyse the potential of using a simulation approach in modelling energy systems at the urban scale.

Having clarified what simulating an urban energy system means, the authors reviewed 40 different open-source models mainly based on the list provided by the Open Energy Modelling Initiative (openmod) [3]. The review considered three main aspects. First, the sectors covered by the model. Second, the type of model, i.e., optimization or simulation, as for the definition provided in Section 2. Last, whether the model allows simulating energy systems at the urban scale.

The review focused on clustering those tools that allow modelling multiple energy vectors (at least electricity and heat ones) at the urban scale. Table 1 shows the list of the models that match these criteria. The complete list of investigated models is available in Table A1 in Appendix A.

**Table 1.** List of models that allow modelling of multiple sectors and at the urban scale.

| Model | Sectors * | Model Type | Urban Scale |
|---|---|---|---|
| Calliope [6] | User-dependent | Optimization | Y |
| OMEGAlpes [18] | Electricity, Heat | Optimization | Y |
| Oemof [19] | El., Heat, Transport | Optimization, Simulation | Y |
| PyPSA [20] | El., Heat, Transport | Optimization, Simulation | Y |
| REopt [21] | Electricity, Heat | Optimization | Y |
| URBS [22] | User-dependent | Optimization | Y |
| CEA [23] | Electricity, Heat | Optimization, Simulation | Y |
| Backbone [24] | All | Optimization | Y |

* Electricity is abbreviated as "El.".

The model type column in Table 1 presents the entries originally provided by openmod. However, considering the definition of simulation given in Section 2, none of the models listed allow actual simulations to be performed. A possible explanation of this finding is to be found in the generic use of the word simulation. Indeed, it is often used as a general term to indicate a standard run of any model. However, in this work, the authors considered simulation as that model type that allows the analysis of an energy system without looking for an optimal solution.

Thus, despite the existence of well-known simulation tools for the analysis of urban energy systems (e.g., EnergyPLAN [25], HOMER Energy [26]), to the authors' knowledge, there is a lack of open-source simulation tools meeting the criteria aforementioned. These criteria include the provided definition of simulation model and the possibility of performing analysis considering multiple sectors at the urban scale. This absence suggests a niche for the development of such models in the field of energy system modelling.

## 3. Methodology

This section first presents the model developed and then focuses on the methodology applied to perform the comparison with Calliope [6], an existing optimization tool. The Multi Energy System Simulator tool (MESS) is a modular, bottom-up, multi-node model that allows the investigation of non optimal solutions by simulating the energy system.

In light of the already mentioned challenges of modelling energy systems ([4,5], MESS) was developed with a set of design goals in mind and was deeply inspired by Calliope [6]. This choice was made to attempt to mitigate the consequences of an additional model in the literature and to improve the interoperability among multiple models.

The main design goals in the development of MESS are: *(i)* the model was built keeping in mind urban level analyses, while maintaining a certain flexibility in terms of spatial resolution; *(ii)* it should be possible to use the model without the need of coding but just by writing human-readable configuration files; *(iii)* the model should be able to perform analyses on systems composed by multiple energy carriers (e.g., electricity, heat and fuels); *(iv)* the model should have a flexible approach to temporal resolution and timeseries; and *(v)* a free and open- source energy system model written in Julia [27].

The structure of this section is the following:

- Section 3.1 presents the functioning of MESS in detail. Explaining the rationale behind the tool's functioning is essential to improve the clarity of the model proposed. Additionally, it increases transparency and allows future users to better understand the tool's characteristics. This subsection includes two additional sub-subsections.

    – Section 3.1.1 presents the configuration files required to run the model.
    – Section 3.1.2 presents, in detail, the structure and architecture of MESS.

- Section 3.2 introduces the comparison performed between MESS and Calliope in order to depict the potential of the simulation approach compared to the optimization one and describes the case study considered for the comparison.

### 3.1. How Does MESS Work?

In order to use MESS, the user has to set up three configuration files to define the system in analysis and the modelling options. The input files are written in YAML to ensure readability and allow the user to intuitively interpret them. Once this step is done, it is possible to run the model by using the Julia REPL.

Additionally, MESS offers a library of predefined technologies to be included by the user in the model. Each technology is part of a group of technologies that show similar behaviour in terms of energy fluxes. This categorization was taken from Calliope and, as in Calliope, the groups are called parents. MESS has six parents: *demand*, *supply*, *supply_grid*, *conversion*, *conversion_plus* and *storage*. A comparison between the parent categories used in Calliope and MESS is given in Table 2.

**Table 2.** Parent technology groups— Comparison between Calliope and MESS.

| Parent | Calliope | MESS | Description |
|---|---|---|---|
| *demand* | Yes | Yes | Energy demand for the defined carrier |
| *supply* | Yes | Yes | Supplies energy to a carrier |
| *supply_plus* | Yes | No | As supply, with additional constraints |
| *supply_grid* | No | Yes | As supply, energy from national grid |
| *storage* | Yes | Yes | Stores energy |
| *transmission* | Yes | No | Transmits energy from one location to another |
| *conversion* | Yes | Yes | Converts energy, one carrier to another |
| *conversion_plus* | Yes | Yes | Converts energy, N carriers to M carriers |
| Total | 7 | 6 | |

In summary, MESS has one parent technology less than Calliope . In particular, this is the result of not considering *supply_plus* and *transmission* parents but considering the

*supply_grid* one. Looking at the parents category more in detail, *demand* technologies represent energy sinks. The energy carrier to be considered must be defined, and a Comma-Separated Values (CSV) file detailing the demand for that carrier at each timestep is required. Technologies with *supply* as a parent represent energy sources.

Renewable energy sources, such as solar photovoltaic or wind turbines, are the most evident examples for this parent. The carrier considered and technology-specific parameters should be defined. Different modelling options might be considered for each technology. Technologies belonging to the *supply_grid* parent represent energy sources from distribution grids not modelled in the analysis, as, for example, national distribution grids or district heating grids. The energy carrier of the energy source must be defined in this case as well.

Then, *conversion* technologies are defined by a single carrier in and a single carrier out (e.g., natural gas-fed boilers), while *conversion_plus* technologies are defined by multiple energy carriers in and/or out (e.g., combined heat and power technologies). Both categories require the definition of technology-specific parameters. Finally, *storage* technologies are defined by the same carrier in and out and, depending on the state of charge and the energy balance, might act as energy sink or energy source.

### 3.1.1. Input Files

The general configuration parameters for the simulation are set in a specific file named *model_specs.yaml*. This allows the user to define the name of the model, the timespan and timestep to be used, as well as if the local electricity network is to be solved and the type of solver to be used. The *techs.yaml* file is used to define and set the input parameters of all different technologies that might be included in the model. Each technology is defined by three subsets of parameters: essentials, constraints and monetary, plus the priority index.

In the essentials subset, the fundamental parameters are to be declared, such as the user-defined technology name, the colour to be used for plotting, the parent and input and/or output carriers. The constraints subset contains the parameters used to set the technical characteristics of the technologies, and the ones to be specified are technology dependent. In the monetary category, costs related to the technology are to be defined, such as the Capital Expenditures (CAPEX), the Operational Expenditures (OPEX), interest rate etc.

The priority parameter is an integer input that sets the priority of each technology i.e., the order in which technologies are to be called by the solver, hence, allowing the user to define different ways of solving the model. The *locations.yaml* file describes the nodes composing the network to be studied and which technologies each node hosts. For each technology, additional node specific data can be set, as installed capacity or timeseries files (e.g., demand curves and capacity factor series), or specific parameters can be superscripted on the general ones defined in the *techs.yaml* file.

In addition to the configuration files, input files might be needed for demand profiles, non dispatchable power sources generation profiles, energy prices etc.

### 3.1.2. MESS Structure

MESS is divided in four major steps, which are: (1) Pre-processor, (2) Core, (3) Post-processor and (4) Plotting. Figure 1 summarizes these four steps and their main functions.

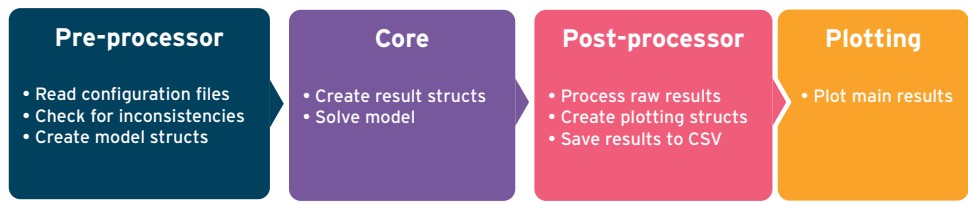

**Figure 1.** Four steps of MESS.

Pre-Processor

In the pre-processor stage, all the modules required for the execution of the following steps are loaded. This includes loading the exceptions and structures modules. The former module contains all the exceptions that might arise in the program, while the latter contains all the data structures used. Some of the structures here defined contain the constraints allowed per each category of technology (or, following MESS's and Calliope's terminology, per each parent) as well as the structures that defines the model characteristics. All this information is then combined with the input files in order to create the model structure to be used in the core module.

Core

In the core stage, the model is solved. Solving the model is a three-step process. In the first step, the single locations are solved at each timestep. In the second step, the solutions of each location are considered together, and the local network is solved. Finally, in the third step, the details of the exchanges with higher level grids (i.e., national grid) are defined. Figure 2 shows a schematic diagram representing the functioning of this phase in MESS.

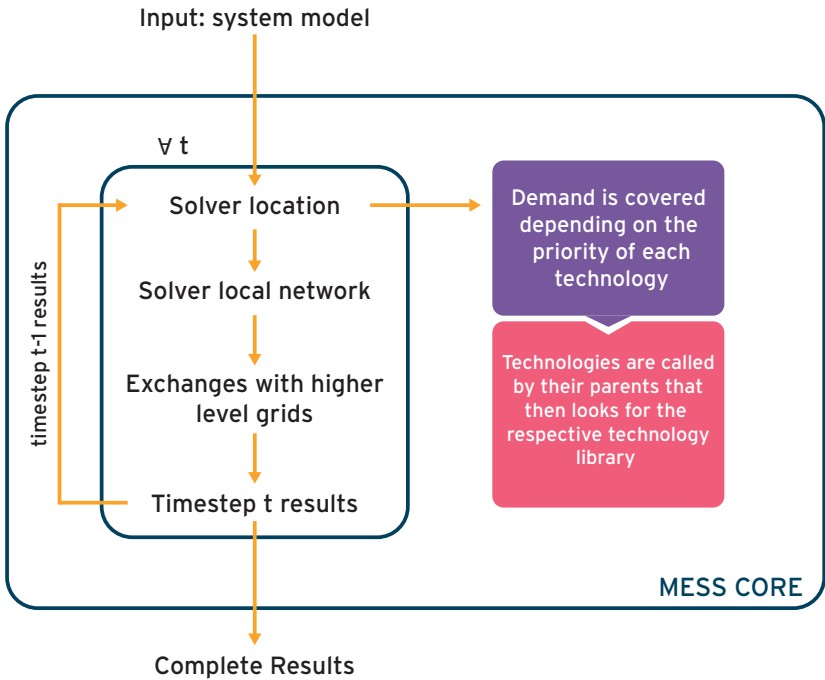

**Figure 2.** Functioning of the core phase of MESS.

As briefly introduced, the first step solves each location at each timestep. The energy balance is initialized at zero at each timestep and is progressively updated while the technologies are solved in each location. Generally speaking, demands are the first to be added to the balance, and then the different technologies are used to cover the demand based on their priority. This means that the technologies with the highest priority (lower index value) are the first to be used to cover the demand.

Once the highest priority technology is called by the solver, it proceeds searching for the second highest one and so on, until the demand is covered. In this phase, using this approach means that non dispatchable renewable energy technologies might lead to an overproduction of energy. This solving strategy adds the possibility of considering counter-intuitive control strategies for each location, expanding the range of scenarios that can be defined and investigated by the user; nonetheless, it should be used with caution, since it might lead to unrealistic behaviours.

Once all locations have been individually solved, the local network is considered in the second step of the core phase. If there are imbalances in the single locations and the option of considering the local network has been considered, the local grid is solved. In this second step, the solver does a simple summation of the positive and negative imbalances of the single locations estimating the amount of energy exchanged. In the third and final step of the core phase, the exchanges with higher level grids (i.e., national grid) are defined.

Post-Processor and Plotting

After having solved the model in the core stage, the generated solution is processed by the post-processor. The objectives of this step are multiple. It allows the processing of data to obtain aggregated indicators of performance, looking at the whole timespan considered and not only at the timestep resolution (e.g., hourly) imposed. At the same time, it allows the processing of the data to proceed with the plotting and to save the results in CSV files that can be used by the user to perform further analyses.

The plotting phase is considered as a separate step from the post-processing one, even though the two are highly interlinked. Plotting in MESS is handled using the PlotlyJS package [28], which creates interactive HTML files that allow the user to analyse the generated plots by zooming in and out and highlighting the single values navigating on the plot. At the current status, the plotting phase automatically generates two different kind of plots. The first one shows the overall results at each timestep, for each location, in terms of the electricity, heat and gas balance. The second one uses the aggregated results for each location to show how the demand is covered in percentage by each technology available in the different locations.

### 3.2. MESS vs. Calliope

Developing and presenting a new model requires a comparison with an existing tool in order to identify its peculiar characteristics and different usage purposes with respect to a renowned standard. Therefore, a comparison between MESS and Calliope [6] was conducted. Calliope is an energy system model that allows to investigate energy systems with high spatial and temporal resolution. This systemt permits the analysis of different scenarios from the urban scale to countries.

The choice of Calliope as a benchmark model was made for several reasons. Calliope has proven to be a largely utilised tool, with high standards of code testing and with an approach that is both user friendly (since no coding is required by the user) and rigorous. At the same time, Calliope is an optimizer, and comparing MESS with it allows us to evaluate the potential of simulating against optimizing a system. Indeed, as mentioned by [12], these two approaches have different strengths and purposes.

Optimization tends to be more indicated for bottom-up models with a high level of technical details and for use by planners and engineers. Nonetheless, due to its characteristics and the long computational times that are usually required, it might show limitations in certain applications. Using a simulation approach results in lower computational times—due to its simpler approach—and might allow a more dynamic and productive interaction with policy makers.

Calliope offers three different modelling options: *(i)* Planning mode allows an investment decision analysis to find the optimal configuration of a system in terms of the installed capacity via the minimization of an objective function.

*(ii)* The operational or dispatch mode is meant to perform an optimization on the economic dispatch of the model. In this case, the installed capacities of the different technologies are fixed, and the model finds the optimal way of satisfying the demand while minimising the objective function.

Last, *(iii)* SPORES mode allows the investigation of sub-optimal solutions around the optimal one. In this work, the first two modes were employed, while the SPORES one was not considered.

The comparison between MESS and Calliope was performed following the steps illustrated in Figure 3.

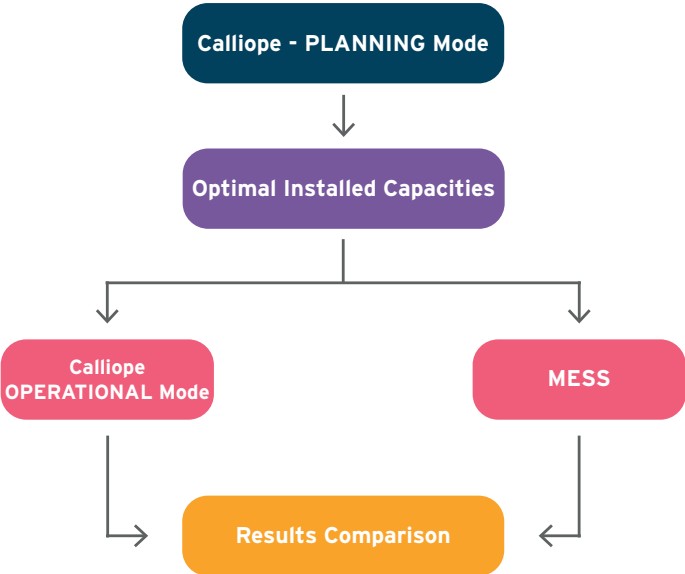

**Figure 3.** Schematic of the steps performed in the comparison between MESS and Calliope.

First, using the planning mode of Calliope allowed us to obtain the optimal level of installed capacities of the different components. The timespan considered (i.e., the horizon used in the run) was one year at an hourly resolution. Then, the operational mode of Calliope and MESS were run with the given installed capacities obtained from running Calliope in planning mode. Afterward, the analysis was focused on comparing the results obtained from the two models.

The comparison of the two models was performed through a case study composed of three locations. The demand profiles considered were obtained from the consumption data of three monitored multi-apartment buildings from the Sinfonia Project [29] in Bolzano, Italy. The load profile of the photovoltaic (PV) panels was obtained from the Renewable.ninja website [30,31] setting Bolzano as the location.

According to the MESS network simulation capabilities, all locations were considered able to exchange electricity with the others. A different mix of technologies was considered for each location. Figure 4 shows the case study considered. In particular, the figure shows the three locations considered, *X1,X2* and *X3*, with the respective demand and technologies. As an example, location *X1* is characterised by both power and heat demands, the connection to the national grid (Supply Grid) and to the District Heating network, by the presence of a combined heat and power (CHP) unit and by a PV unit. All three locations are connected to each other through electrical power lines.

All the characteristics of the technologies considered in MESS and how they are modelled are available in the MESS repository under this file (https://gitlab.inf.unibz.it/URS/MESS/mess/-/blob/da7667912326703f61faf42be66d23cec2874b63/src/core/technologies.jl, accessed on 1 September 2021).

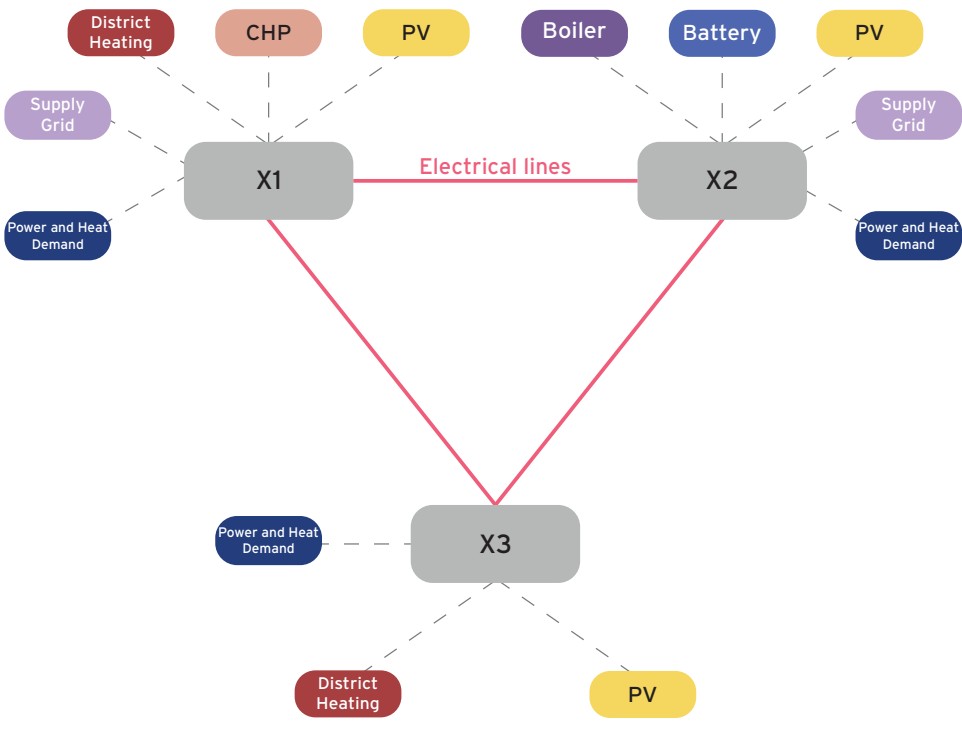

**Figure 4.** Case study example.

## 4. Results

In this section, the results obtained from Calliope and MESS are reported. Calliope was used both in planning (Section 4.1) and operational modes, while MESS was compared to the results obtained from the latter approach (Section 4.2). Table 3 lists the execution times of the three simulations, considering a three node system for a timespan of 1 year and a hourly timestep, which were conducted on a Linux machine running Ubuntu 18.04 with the 15 GB of RAM and a Intel® Core i7-8565U CPU @ 1.8 GHz 64 bit. The time reported in the table were derived from a single-run and include all the phases from pre-processing to plotting. Runs were carried out using the three location case study presented in Figure 4.

**Table 3.** Comparison of the execution times for the three simulations.

| Model | Execution Time |
|---|---|
| Calliope—Planning | ∼45 min |
| Calliope—Operational | ∼11 min |
| MESS | ∼1 min |

The absolute speed achieved does not represent the central aspect of this result. However, having a faster tool allows performing multiple scenarios analysis, making it possible to use it in a participatory process of decision making.

### 4.1. Calliope—Planning Mode

Planning mode was employed to obtain the optimal capacities to be used as inputs for each technology for the following simulations. The obtained capacities are shown in Table 4. Given the costs imposed, the optimized results would tend not to include photovoltaic panels and batteries in the technology mix; hence, a lower bound on their capacity to be installed was imposed. Given the constraints, locations X1, X2 and X3 resulted as having, respectively, 5.0, 10.0 and 7.0 kW installed of PV, while an energy storage system of 5.0 kWh was imposed in location X2 as well. Locations X1 and X3 mainly rely on district heating to cover their thermal demands, with a minor contribution from CHP in X1, while a boiler unit supplies thermal energy to X2.

**Table 4.** The Calliope planning mode results, used as inputs for the Calliope operational mode and MESS.

| Location | Technology | Installed Capacity |
|----------|------------|--------------------|
| X1 | CHP | 9.1 kW |
| X1 | District heating | 183.1 kW |
| X1 | PV | 5.0 kW |
| X1 | Supply gas | 22.5 kW |
| X1 | Supply grid power | 20.9 kW |
| X2 | Battery | 5.0 kWh |
| X2 | Boiler | 50.8 kW |
| X2 | PV | 10.0 kW |
| X2 | Supply gas | 59.8 kW |
| X3 | District heating | 131.4 kW |
| X3 | PV | 7.0 kW |

*4.2. Calliope—Operational Mode and MESS*

Given the capacities obtained from the investment planning optimization, the operational mode in Calliope and the MESS simulation were run. In this subsection, the results obtained are presented. In particular:

- The first section presents the aggregated results at an annual level to identify differences between the two approaches at a macro level.
- The second section shows the results on a monthly-based scale to depict the differences for the single months and, hence, the seasonality, arising from how the two models work.
- The third section presents the results for four representative weeks (one each for winter, spring, summer and autumn) at an hourly level. In this way, it is possible to have an overview of the differences between the two models in solving the hourly balance.

4.2.1. Annual Aggregated Results

Figures 5 and 6 show the results aggregated for the whole timespan considered (8760 h, 1 year). Each bar in Figure 5 shows the total amount of electricity obtained from each technology: blue bars represent Calliope's results, while the green ones represent MESS. The energy produced by the PV panels is exactly the same in all three locations; this should not be surprising given the straightforward functioning of a non-dispatchable technology and the simple models employed. Differences can be noted both for the CHP ($\sim$16%) in location X1 and for the battery in location X2 ($\sim$35%).

In the former case, such a difference might be ascribed to the CHP producing electricity not only for location X1 but also for the other locations in the case of Calliope. Indeed, this possibility is yet to be implemented in MESS: dispatchable technologies can only be controlled by the demand of the location where the technology is installed. In the case of the battery, in Calliope, its usage depends on a economic optimization of the system as a whole, while in MESS it only tends to maximize the electricity self-consumption of the location where it is installed.

Finally, the most evident difference is in the electrical energy imported from the national grid. Looking at Calliope's results, electricity is only imported in location X1: this is because the connection to the grid is placed there, and electricity is then distributed to X2 and X3 from X1. In the case of MESS, no electricity is imported in X1 since the location is self-sufficient, while substantial imports are present in X2 and X3, since all locations are supposed to be connected to the grid.

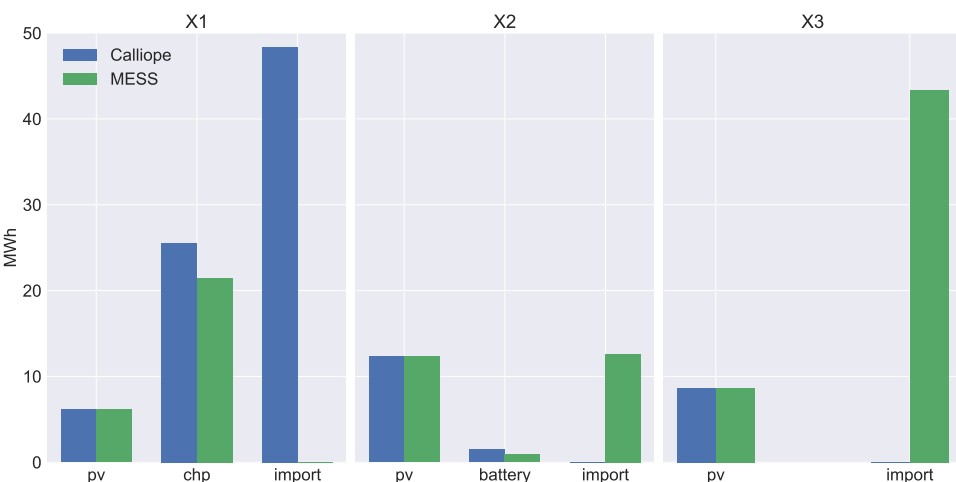

**Figure 5.** Annual electricity per technology source—Calliope and MESS comparison.

Similar considerations can be made for Figure 6. The boiler in X2 and district heating in X3 are the only heat sources for their locations, and the results obtained from Calliope and MESS match completely. The differences highlighted for the CHP in the electricity case have repercussions on the heating part for location X1 as well. The CHP is operated in the electrical load following mode, hence the higher quantity of electricity generated in Calliope's solution translates in a higher production of heat as well, which is compensated by MESS with an higher quantity of heat purchased from the district heating grid.

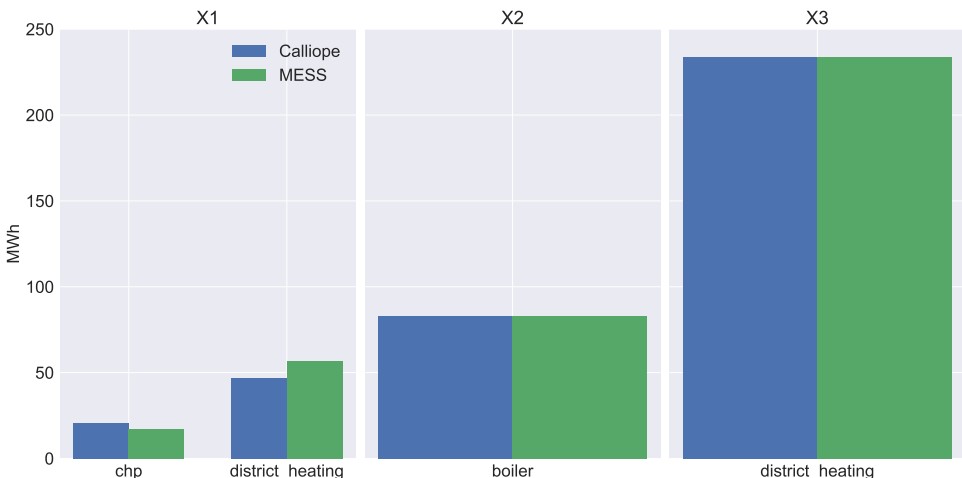

**Figure 6.** Annual heating per technology source—Calliope and MESS comparison.

### 4.2.2. Monthly Aggregated Results

Monthly aggregated results are here shown for location X1. The same results for location X2 and X3 can be found in Appendix B. Figure 7 shows the monthly amount of energy derived from different technologies for Calliope (left-hand side graph) and MESS (right-hand side graph). As seen in Figure 4, the differences between the two models are in how the CHP works and the reliance on imported energy. Calliope shows a greater utilization of CHP in winter months, while there is a heavier reliance on electricity imports in the summer.

This could be ascribed to the higher thermal demand of the winter months: in that case, it would make more economic sense to have the CHP running rather then buying electricity from the grid, since the CHP could provide both electrical and thermal energy. On the other hand, MESS shows a more regular behaviour of the CHP throughout the year.

As seen in the previous paragraph, the CHP in MESS works in a electrical load following mode; hence, its behaviour is only dictated by the electricity demand and the PV production, resulting in a more even behaviour. Moreover, no electricity is imported from the grid, since the CHP size is enough to cover, together with the PV panels, the electricity demand.

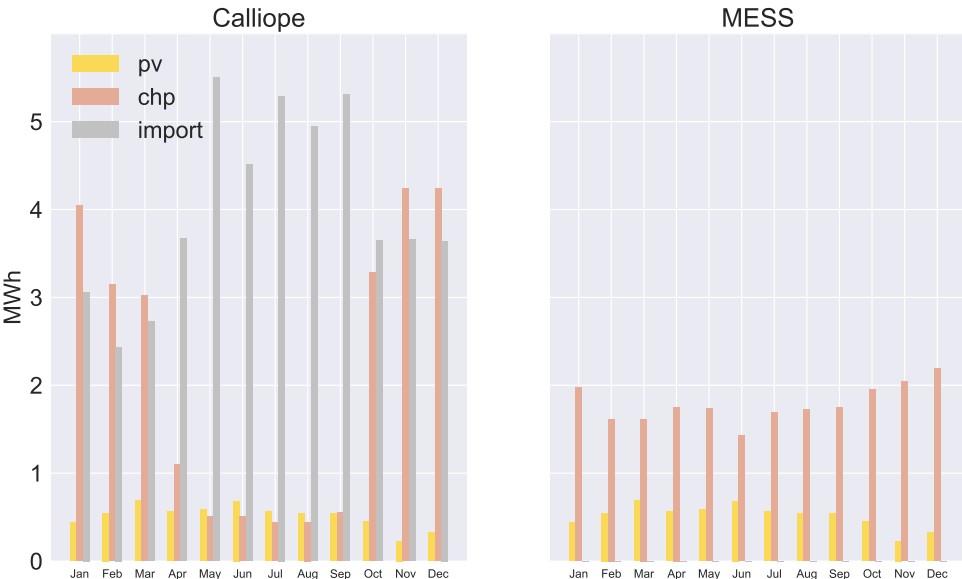

**Figure 7.** Monthly electricity production per technology source: Location X1—Calliope and MESS comparison.

The graphs in Figure 8 confirm what was said about Figure 6. Indeed, the results obtained with Calliope show a higher heat production from the CHP in the winter and a way lower production in the summer. Instead, MESS relies more heavily on the district heating in colder months and has an excess production of heat in the warmer ones—heat that is, hence, discarded. Details of the monthly behaviour for all the locations are presented in Appendix B.

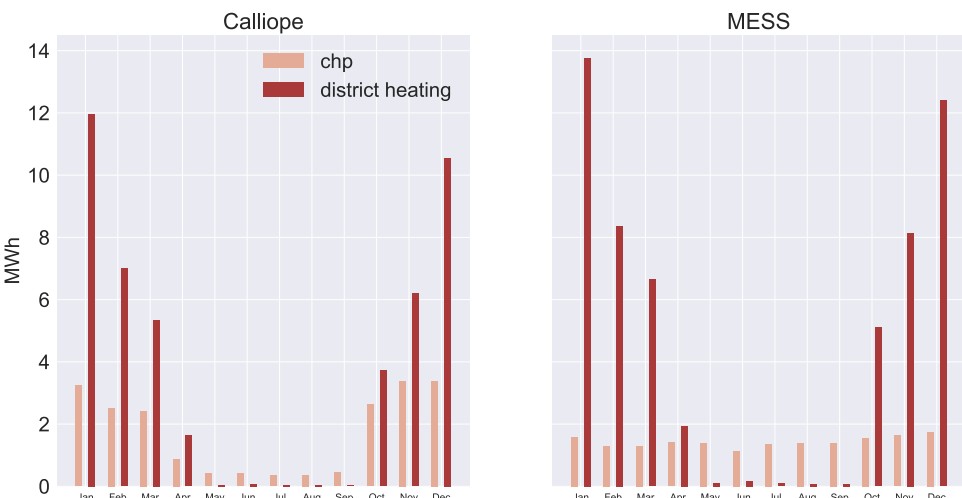

**Figure 8.** Monthly heating per source: Location X1—Calliope and MESS comparison.

### 4.2.3. Hourly Results—Typical Weeks

Finally, the results obtained from the two modelling tools are shown on an hourly basis for four representative weeks of the year. Figure 9 shows the results obtained via

Calliope for a week in winter, spring, summer and autumn, while Figure 10 shows the same results for MESS.

Looking at Figure 9, it is possible to notice a similar behaviour for the winter and autumn weeks and for the spring and summer ones. The main difference between the two pairs is the behaviour of the CHP. In the colder seasons, the CHP has a major role, since it allows coverage of both the electrical and thermal demands as seen also in the previous paragraphs.

The reliance on the grid is much heavier in the warmer seasons, since the contribution of the CHP is almost negligible. The electricity demand is always exceeded by the electrical energy produced or imported from the grid. This is because location X1 acts as a connection point for all three locations to the national grid. In winter and autumn, the CHP tends to reach its peak production, and the remaining electricity demand from location X2 and X3 is covered by buying electricity from the grid. In spring and summer, since the thermal demand is lower, it makes more economic sense to buy electricity from the grid, and the CHP is used much less.

Another thing worth noticing is the unmet demand at the beginning of the spring week. In this case, the electricity demand in X1 is actually met; however, not from a combination of the technologies seen so far but from an excess of PV electricity from the other locations, since it happens in the central hours of the day.

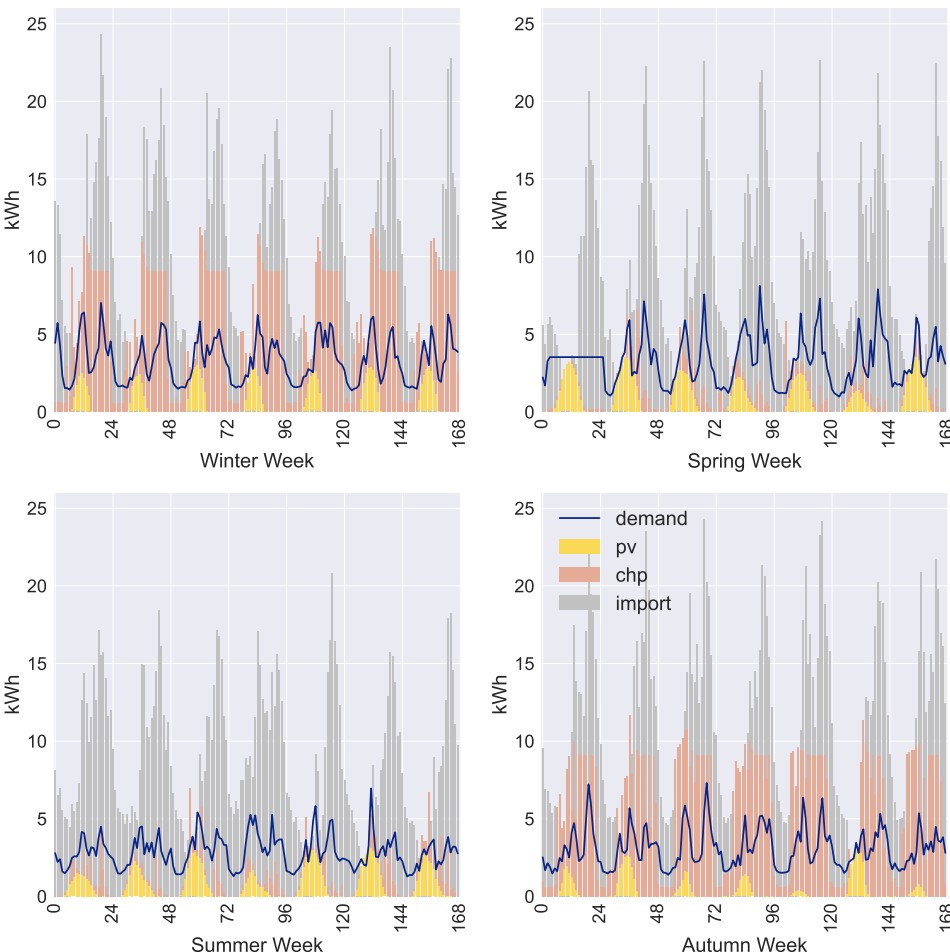

**Figure 9.** Hourly electricity production of four representative weeks: Location X1—Calliope.

In the case of MESS, the interpretation of the results shown in Figure 10 is more straightforward, since each location tends to be more independent and in general less reliance is made on the local grid. In of the considered weeks, the demand is completely

satisfied by the combination of PV panels and CHP. Priority is given to the non-dispatchable electricity produced by the PV panels, while the CHP covers the remaining demand.

Given a good superposition of production and demand and the size of the solar panels, almost no excess electricity is produced in the analysed weeks, except for a very few hours in the summer. In that case, the excess electricity is exported to the other locations, if required, or otherwise sold to the grid. In Appendix C, it is possible to see the weekly results for the other two locations.

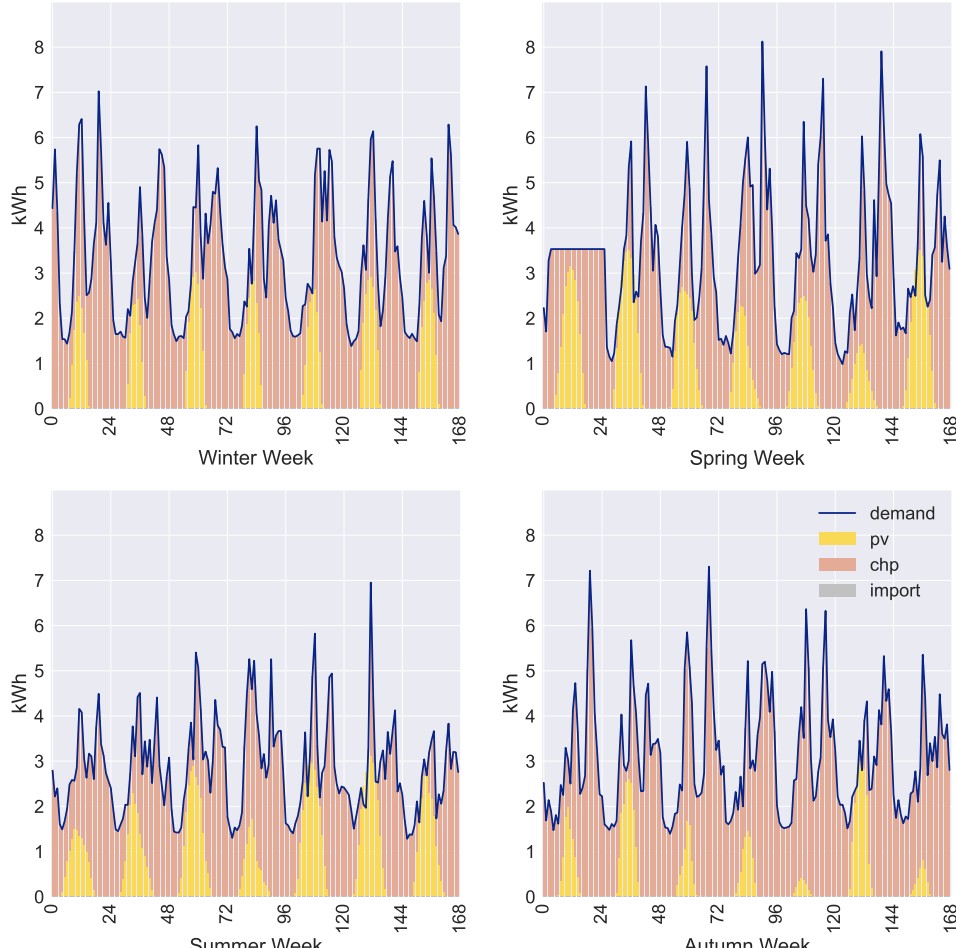

**Figure 10.** Hourly electricity production of four representative weeks: Location X1—MESS.

## 5. Discussion

The analysis of the results made it possible to observe some of the differences between MESS and Calliope. The comparison has shown how these differences derive from different principles and strategies adopted by the two models in solving the system. In Calliope, the optimization aims at minimizing whole system running costs, and this is the principle on which the functioning of each component is based.

MESS, on the other hand, has its own rules to solve each technology, which led to the differences highlighted in the results for components as the CHP or the battery system. The differences are limited not only to how single components are solved but also to how Calliope solves the network with respect to MESS.

Indeed, while MESS gives priority to the self sufficiency of each location, in Calliope technologies can also contribute to cover the demand of other locations—always following the principle of whole system running cost minimization. From the results obtained, and considering the research questions introduced in Section 1.1, the following considerations can be made.

The modest differences at the yearly balance between the two models, denotes how a simplified approach, like the simulation one, depending on the application considered, could provide results of satisfactory precision, while significantly reducing the computational times and offering results that might be easier to be interpret based on predefined logic as possibly defined by the user.

This result is relevant since it suggests that, in some situations, a simulation approach might be the right choice: while an optimization approach is more indicated for investment planning models and macro energy systems analysis, a simulation approach might be more suitable for quick investigations of numerous scenarios on a smaller scale, making it an interesting option for a wider set of stakeholders in their decision process. Shorter execution times, together with an approach that makes it easier to understand the logic behind the model might contribute to making the modelling process more open and inclusive.

Indeed, it might be possible to include the modelling process in meetings as well as workshops and information campaigns to support the design of new policies and energy strategies. In this way, it would be possible to follow a more transparent and participatory approach. Moreover, as suggested by Pfenninger et al. [4], a model run taking seconds instead of minutes could allow modellers to perform uncertainty and sensitivity analyses on numerous parameters.

Regarding the potential of simulating energy systems, another aspect to be considered is that some of the solving principles applied in MESS, despite being simple, are particularly realistic when considering an urban context. To give an example, it might be more likely that, in certain areas, the majority of owners of a battery will tend to use it to store the excess of production from their photovoltaic modules rather than to trade energy with the grid to enhance profits.

In this sense, an optimization approach is less flexible and might make it more difficult to represent non-optimal behaviours. Instead in MESS, using a simulation approach makes it easier to implement different solving strategies that might be closer to the optimized ones, e.g., technology dispatching according to price signals. On the other hand, predefined strategies might enforce faulty logics leading to non-realistic or biased results. Therefore, a combined use of optimization and simulation tools might be a way to explore non-optimal solutions using the optimal one as a reference case.

## 6. Conclusions

In this work, the authors presented a newly developed model, called MESS—Multi Energy System Simulator. We compared it to a benchmark optimization model (Calliope) to investigate the potential of simulating energy systems and the advantages and disadvantages compared to an optimization approach.

The results demonstrated that, despite differences in how the energy demand is covered, mainly due to the logic behind the network solver and the cost minimization approach, the overall yearly results tended to be similar. This outcome reinforced the idea that, despite the simplified approach, it is possible to use a simulation approach to analyse energy systems at an urban scale with satisfactory results. Furthermore, the simplified approach brings advantages in investigating multiple alternatives and scenarios in a relatively short amount of time.

Simpler models might help to address one of the challenges of energy system modelling: the need for a higher level of transparency. Indeed, more transparency could contribute to improve the democratization process of analysing and planning energy systems and, in the end, fostering a fair and just energy transition. Additionally, by developing an open-source simulation tool, the authors aimed at addressing some of the challenges presented by Pfenninger et al. [4] regarding modelling energy systems.

In particular, providing the source code and access to the data used for the simulations improved the transparency and accessibility of the developed tool. Combining this aspect with the clear explanations of the definitions presented in Section 2 improved the proposed

work's reproducibility. Additionally, this also aids in enhancing accessibility for the tool developed.

However, the approach used in this work presents certain shortcomings. The differences in the results at the hourly resolution represent a signal that, if the goal of the modelling is to design a system from a technical perspective, the use of an optimization approach appears to be more indicated. Additionally, comparing the developed tool with other existing tools can provide additional insights into the potential of the simulation approach utilized in this work.

Finally, MESS was developed with a high level of flexibility and modularity in mind. Thus, future areas of research and development will be oriented toward the improvement of the library of technologies, including different modelling options for each technology, toward the integration of the spatial dimension, which is crucial to plan and analyse future energy systems at the urban level and toward the analysis of the effects of different energy policies.

**Author Contributions:** Conceptualization, L.B., P.L., P.Z., C.C. and L.K.; introduction, L.B. and P.L., methodology, L.B. and P.L.; software, L.B., P.L. and P.Z.; formal analysis, L.B. and P.L.; data curation, L.B.; writing—original draft preparation, L.B. and P.L.; writing—review and editing, P.Z., C.C. and L.K.; visualization, L.B. All authors have read and agreed to the published version of the manuscript.

**Funding:** This research was funded by European Union's Horizon 2020 research and innovation programme under the Marie Skłodowska-Curie Actions, Innovative Training Networks, Grant Agreement No 812730 (SMART BEEjS Project).

**Data Availability Statement:** The data used in this work are publicly available in the following git repository (https://gitlab.inf.unibz.it/URS/MESS/mess-energies-mdpi, accessed on 1 September 2021). Here it is possible to find the configuration files used for running models in Calliope (both the Planning and Dispatch one) and MESS.

**Acknowledgments:** The authors thanks the SINFONIA Project for making available the data used for the simulations (Grant Agreement No 609019).

**Conflicts of Interest:** The authors declare no conflict of interest.

## Abbreviations

The following abbreviations are used in this manuscript:

| | |
|---|---|
| CAPEX | Capital Expenditure |
| CHP | Combined Heat and Power |
| CSV | Comma-Separated Value |
| IPCC | Intergovernmental Panel on Climate Change |
| LAU | Local Administrative Unit |
| MESS | Multi Energy System Simulator |
| openmod | Open Energy Modelling |
| OPEX | Operational Expenditure |
| PV | Photovoltaic |
| RES | Renewable Energy Sources |
| RAM | Random-Access Memory |

## Appendix A. Models Review

Table A1. List of models reviewed.

| Model | Sectors * | Math Modeltype | Timeresolution | Georesolution | Urban Scale | Modelling Software |
|---|---|---|---|---|---|---|
| Backbone | All | Optimization | Hour | Depends on user | Y | GAMS |
| Balmorel | El. , Heat | Optimization | Hour | NUTS3 | N | GAMS |
| CAPOW | El. | Simulation | Hour | Zonal | N | PYTHON-PYOMO |
| Calliope | User-dependent | Optimization | Hour | User-dependent | Y | PYTHON-PYOMO |
| DESSTinEE | El. | Simulation | Hour | National | N | EXCEL-VBA |
| DIETER | El. and Sector Coupling | Optimization | Hour | Node | N | GAMS-CPLEX |
| Dispa-SET | El. | Optimization | Hour | NUTS1 | N | PYTHON-PYOMO, GAMS |
| ELMOD | El. , Heat | Optimization | Hour | Network | N | GAMS |
| EMLab-Generation | El. , Carbon | Simulation | Year | Zones | N | JAVA |
| EMMA | El. | Optimization | Hour | Country | N | GAMS |
| ESO-X | El. | Optimization | Hour | Node | N | GAMS-CPLEX |
| Energy Transition Model | El., Heat, Transport | Simulation | Year | Country | N | RUBY-RAILS |
| EnergyNumbers-Balancing | El. | Simulation | Hour | National | N | FORTRAN |
| EnergyRt | | Optimization | | | N | GAMS-CPLEX |
| EnergyScope | El., Heat, Transport | Optimization | Hour | Country | N | GLPK-CPLEX |
| Ficus | El. , Heat | Optimization | 15 Minute | | | PYTHON-PYOMO |
| FlexiGIS | El. | Opti., Simulation | 15 Minute | Urban | Y | |
| Genesys | El. | Opti., Simulation | Hour | EUMENA, 21 regions | N | C++ |
| GridCal | El. | Opti., Simulation | | | | PYTHON |
| MEDEAS | El. , Heat | Other | Year | Global, continents, nations | N | PYTHON |
| NEMO | | Opti., Simulation | Hour | NEM regions | N | PYTHON |
| OMEGAlpes | El. , Heat | Optimization | | | Y | PYTHON |
| OSeMOSYS | All | Optimization | Day | Country | N | PYTHON |
| Oemof | El., Heat, Transport | Opti., Simulation | Hour | Depends on user | Y | PYTHON-PYOMO |
| OnSSET | | Optimization | Multi year | 1 km to 10 km | Y | PYTHON |
| PowNet | El. | Opti., Simulation | Hour | High-voltage substation | N | PYTHON-PYOMO |
| PowerMatcher | | | | | | JAVA |
| PyPSA | El., Heat, Transport | Opti., Simulation | Hour | User dependent | Y | PYTHON-PYOMO |

**Table A1.** *Cont.*

| Model | Sectors * | Math Modeltype | Timeresolution | Georesolution | Urban Scale | Modelling Software |
|---|---|---|---|---|---|---|
| REopt | El. , Heat | Optimization | Hour | Site | Y | JULIA-JuMP |
| Region4FLEX | El. and Sector Coupling | Optimization | 15 Minute | Administrative districts | Y | PYTHON |
| Renpass | El. | Opti., Simulation | Hour | Regional (only DE) or Country. | N | R |
| SIREN | El. | Simulation | Hour | | N | PYTHON |
| SciGRID | El. , Transmission | Simulation | | Nodal resolution | | PYTHON |
| SimSES | El. | Simulation | Minute | | N | MATLAB |
| StELMOD | El. | Optimization | Hour | Nodal resolution | | GAMS |
| Switch | El. | Optimization | Hour | buildings, to continental | Y | PYTHON-PYOMO |
| Temoa | All | Optimization | Multi year | single region | N | PYTHON-PYOMO |
| TransiEnt | El., Heat, Gas | Simulation | Second | Hamburg | N | MODELICA |
| URBS | User-dependent | Optimization | Hour | User-dependent | Y | PYTHON-PYOMO |
| City Energy Analyst | El. , Heat | Optimization, Simulation | Hour | | Y | PYTHON |
| Total | | | | | | 40 |

* Electricity is here abbreviated with "El.".

## Appendix B. Monthly Results

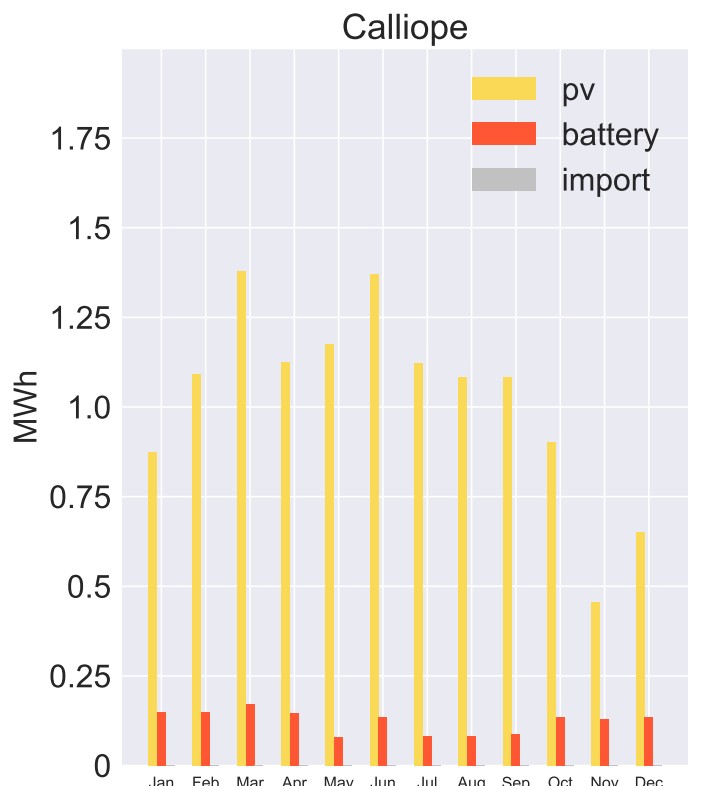
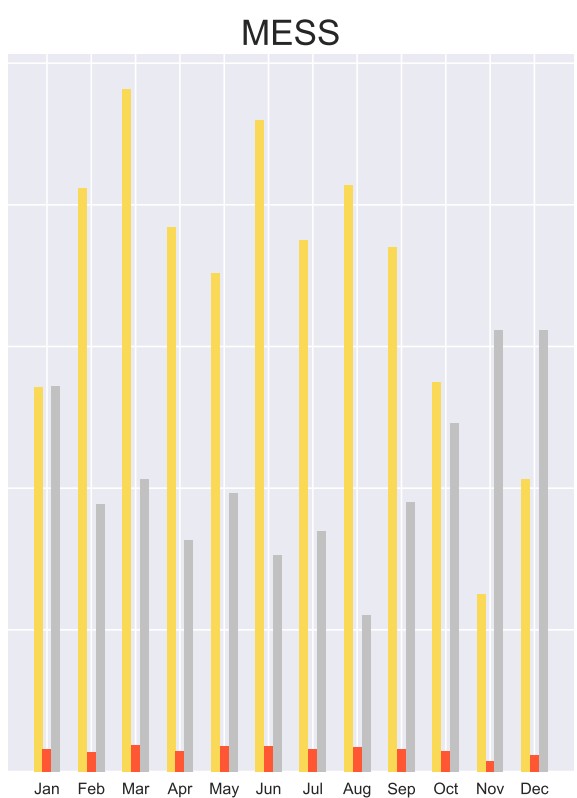

**Figure A1.** Monthly electricity production per technology source: Location X2—Calliope and MESS comparison.

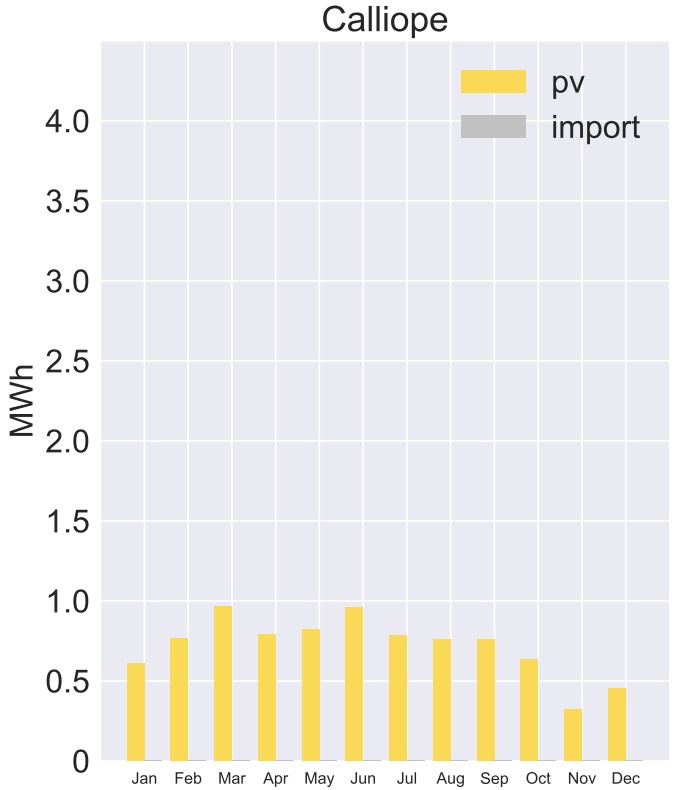
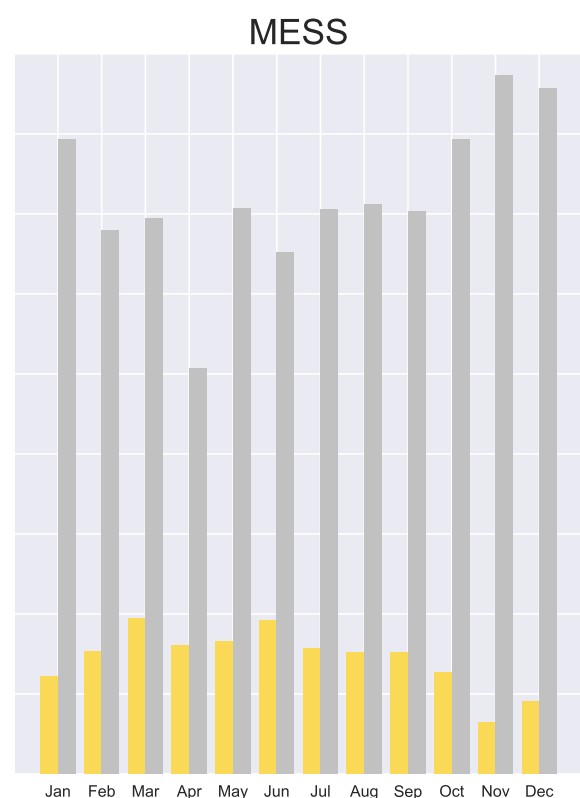

**Figure A2.** Monthly electricity production per technology source: Location X3—Calliope and MESS comparison.

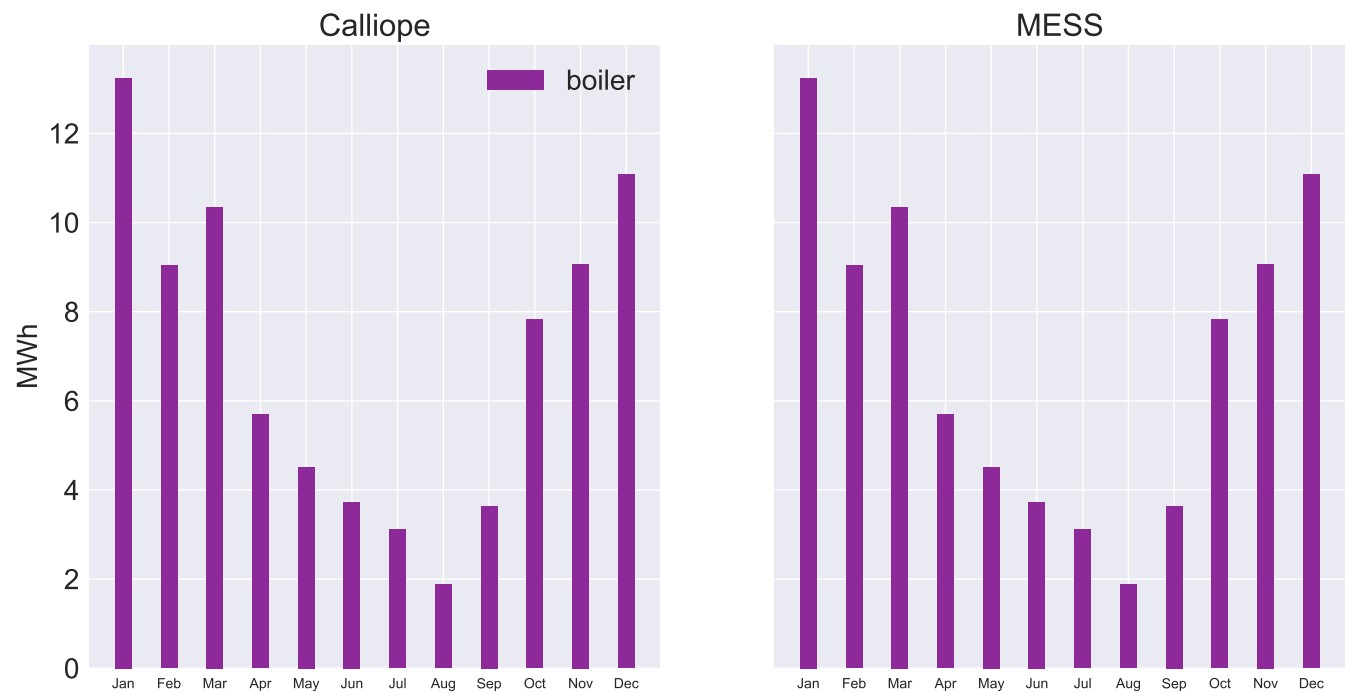

**Figure A3.** Monthly heating per technology source: Location X2—Calliope and MESS comparison.

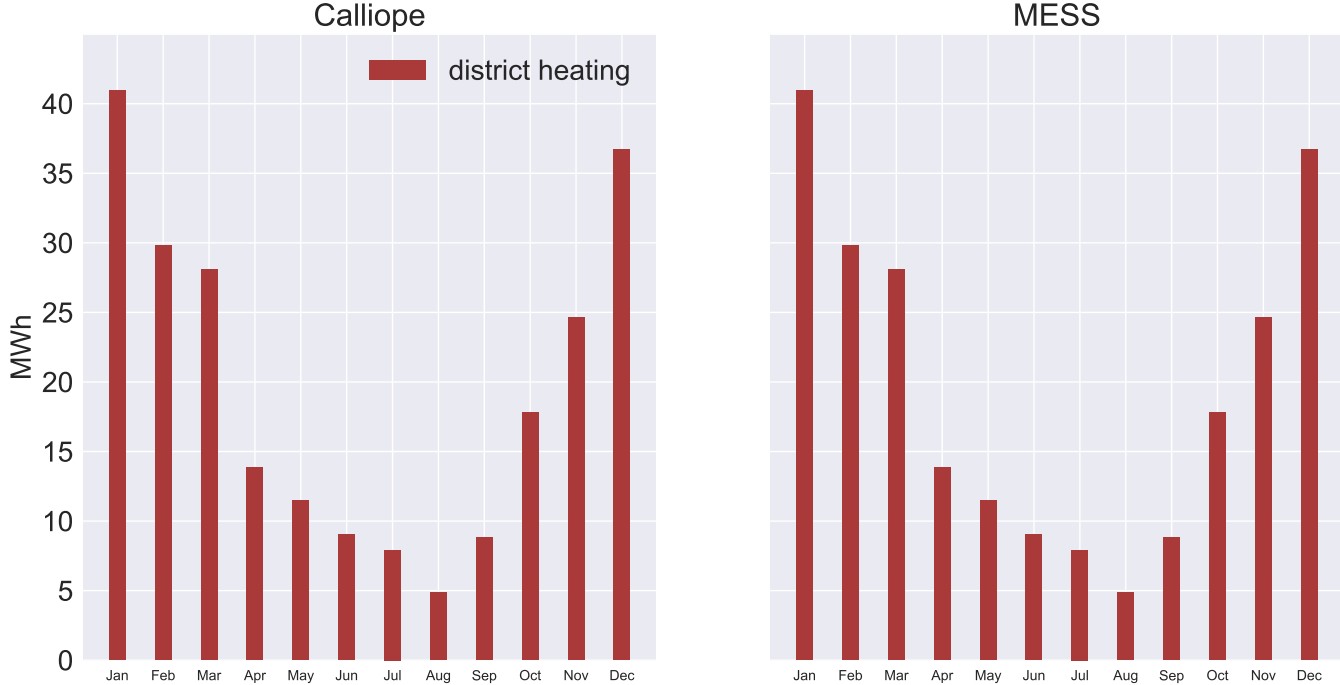

**Figure A4.** Monthly heating per technology source: Location X3—Calliope and MESS comparison.

**Appendix C. Weekly Results**

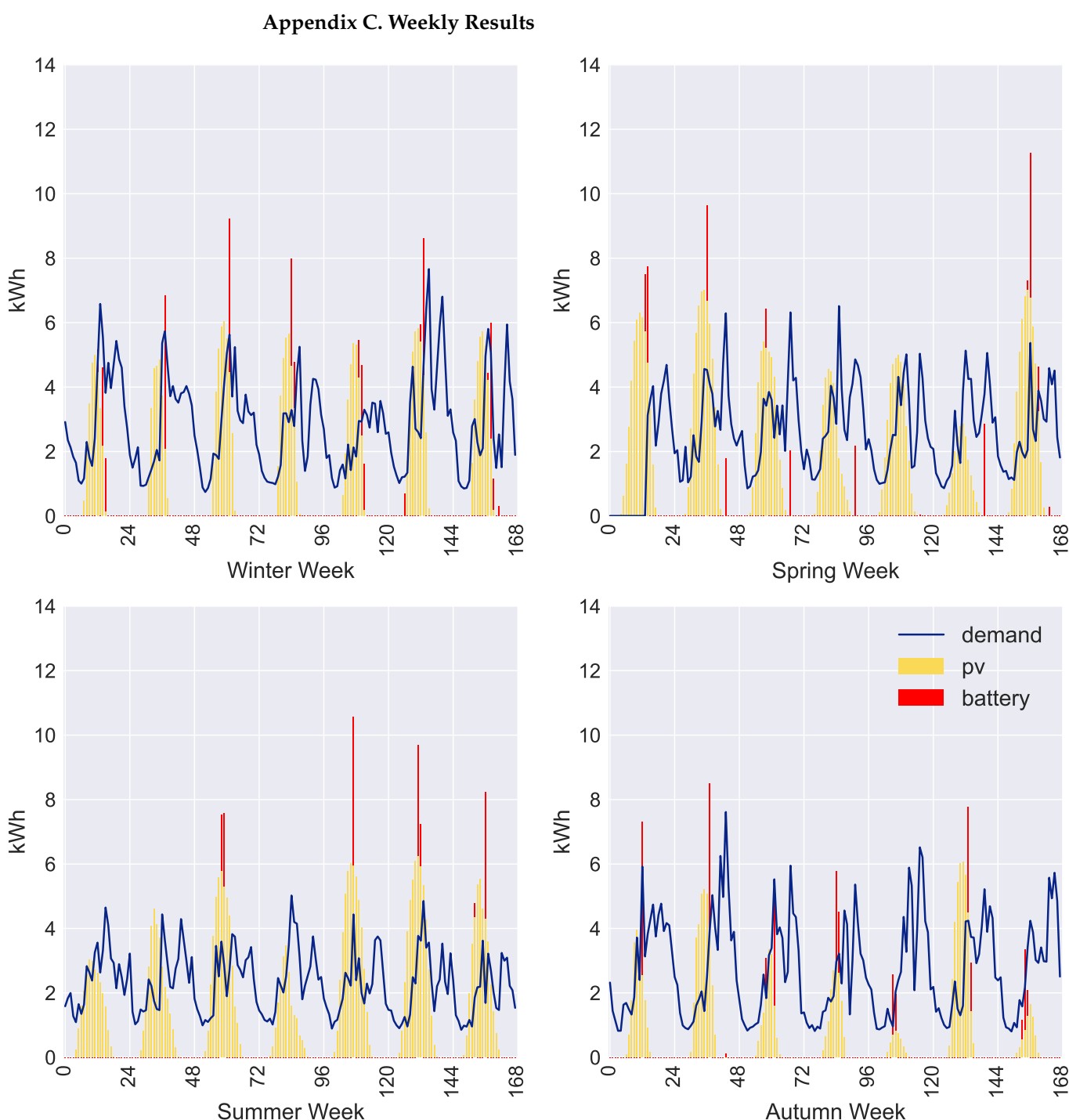

**Figure A5.** Weekly electricity production per technology source: Location X2—Calliope.

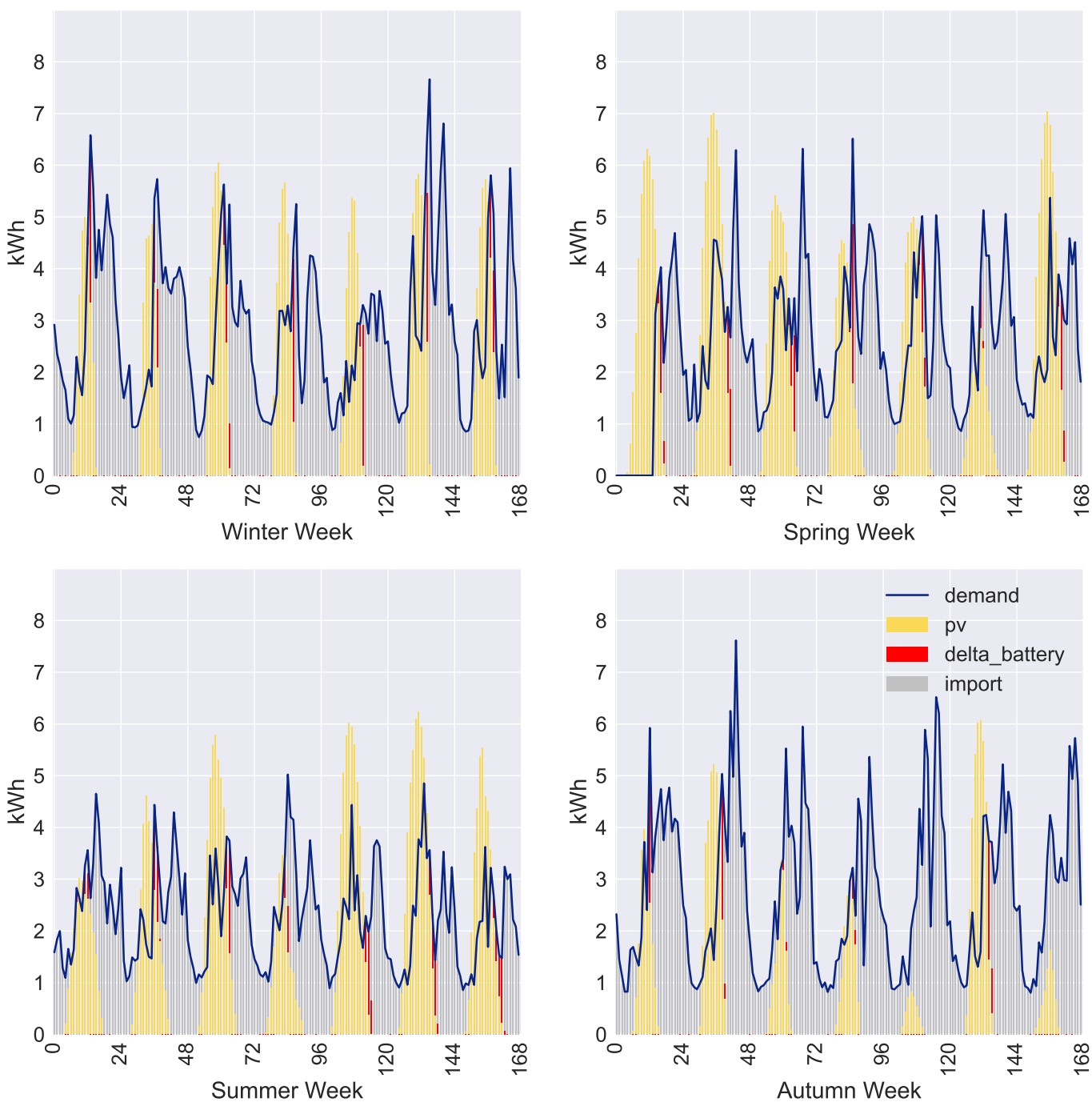

**Figure A6.** Weekly electricity production per technology source: Location X2—MESS.

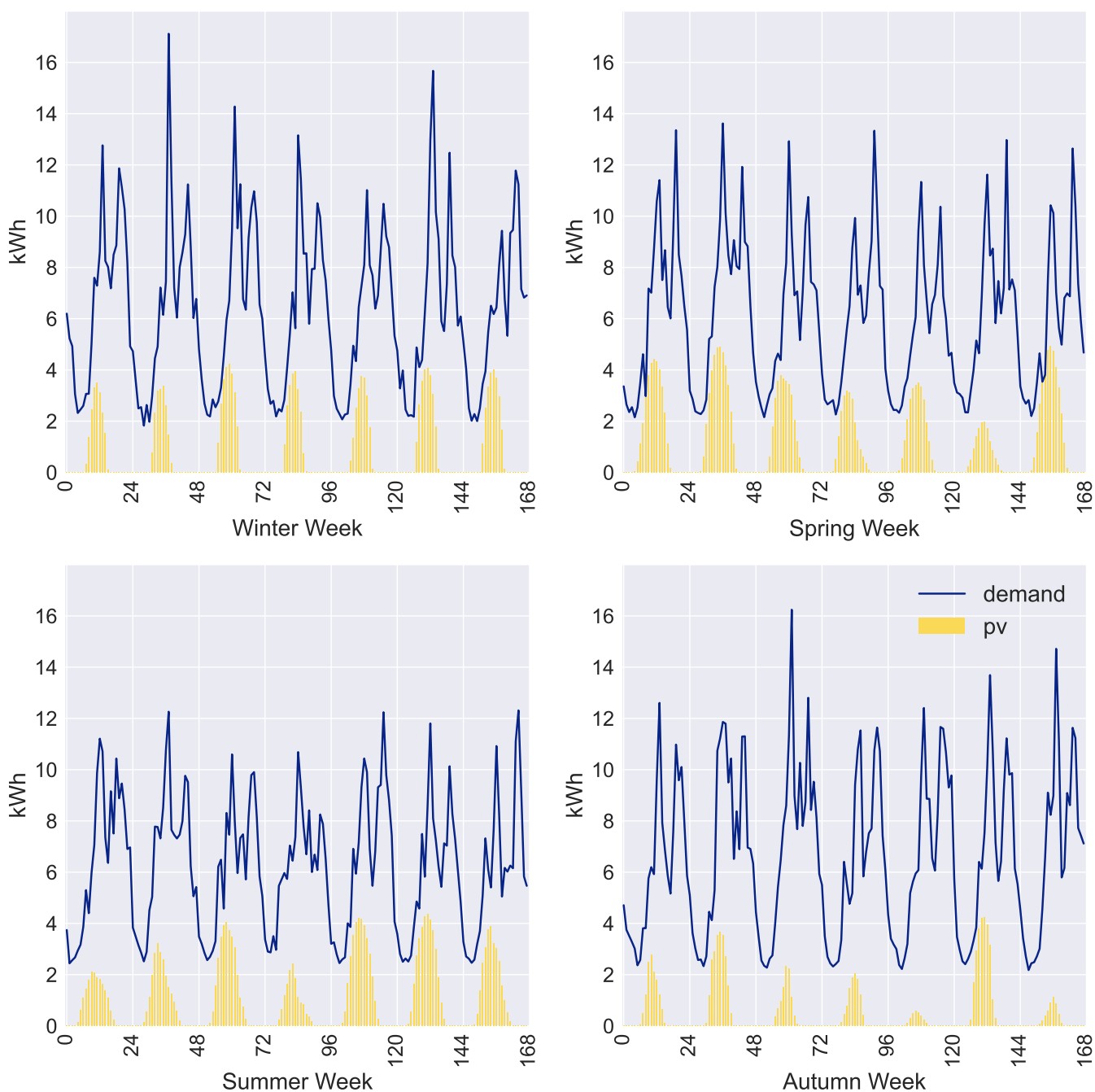

**Figure A7.** Weekly electricity production per technology source: Location X3—Calliope.

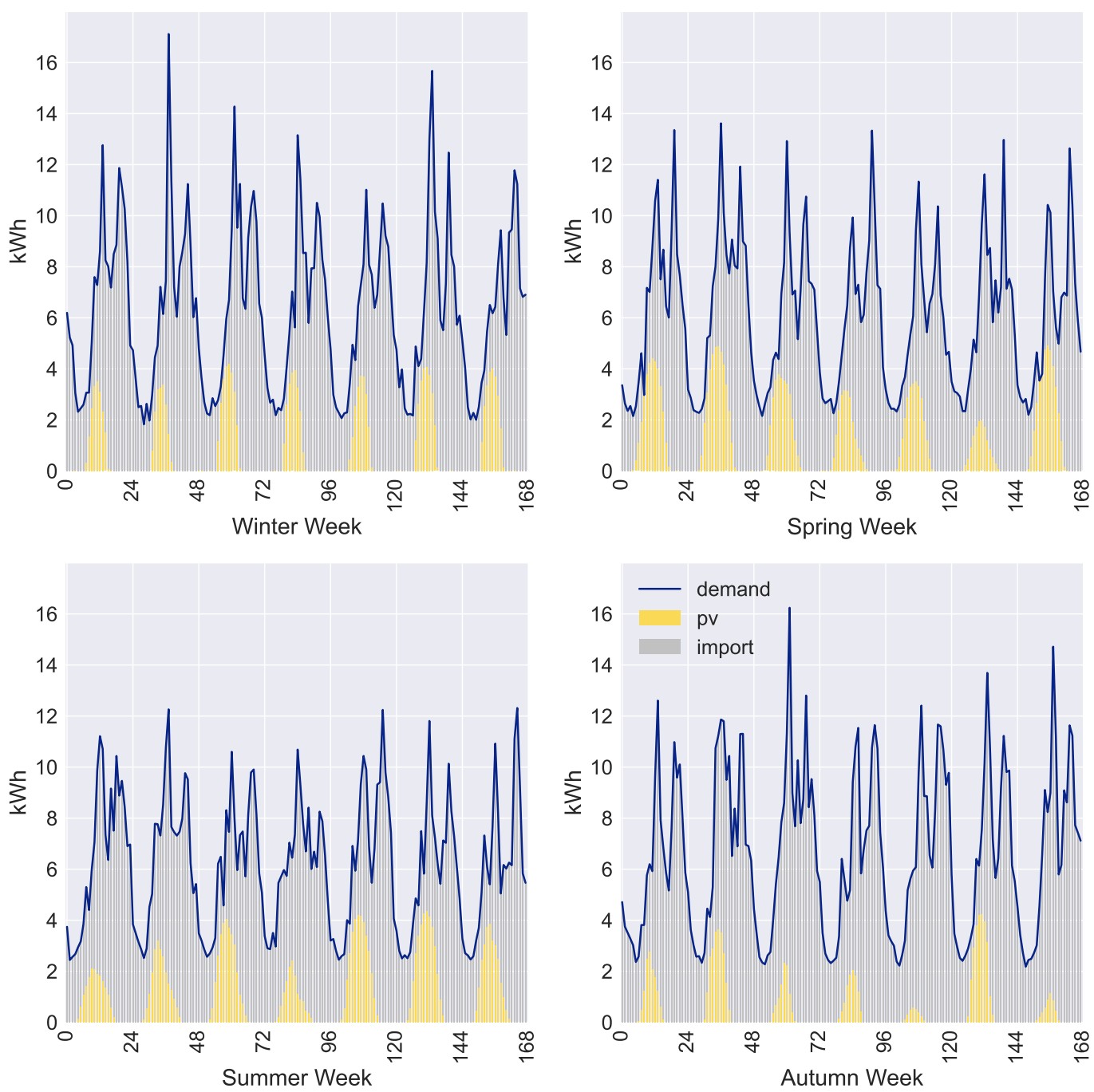

**Figure A8.** Weekly electricity production per technology source: Location X3—MESS.

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
