# Peer review of "The Potential of Simulating Energy Systems: The Multi Energy Systems Simulator Model"

_energies, doi:10.3390/en14185724_

Round 1

Reviewer 1 Report

Dear Authors,

Thank You for the opportunity of reading this article. My generall assessment about this article is positive.

General statements about the article:

-> The article proposed model called MESS - Multi Energy System Simulator and compared it to a benchmark optimization model (Calliope) to investigate the potential of simulating energy systems and the advantages and disadvantages with respect to an optimization approach. Thus the topic and scope of the article is interesting, actual and highly desirable.

-> The article content suite to Energies journal scope.

-> abstract is adequate to article content

-> Organization of the paper is clear and correct.

-> Keywords are correctly proposed.

-> Literature review is based on 31. They are related to article content.

However, I indicated the following elements to revision:

#1

Please revise subsection 1.3 to highlight more the contribution of the paper.

#2

Please extend conclusions with the limitation of the proposed approach

#3

Please also revise the manuscript regarding the personal way of addressing in the text. Please avoid and replace we" or "our" with the impersonal manner of addressing. The text will sound much more professional.

#4

In the article, there is a noticeable number of abbreviations. Thus I recommend adding a list of main abbreviations.

Best regards,

Reviewer

Reviewer 2 Report

The article, "The potential of simulating energy systems: the Multi Energy Systems Simulator model" is intended to present a developed tool called Multi Energy Systems Simulator (MESS) that allows to investigate non optimal solutions by simulating the energy system by a comparison between MESS itself and an optimization model, in order to analyze and highlight the differences between the two approaches, the potentialities of a simulation tool and possible areas for further development.

The main subject of this article, the need of providing full access to the code and the data used trough an open source approach, is important. However, the authors must consider the following major improvements before the article can be accepted and published:

  • General comments:
    • English: Please, check and use the temporal constructions consequently, for example: “will be”? (13)
    • All the used abbreviations written for the 1. time, such as IPCC (36) should be written with their full names and possibly collected and explained at the beginning of the article.
    • Introductions to sections should not start with a space (for example such as 140, 142?) or 157, … and so on.
    • Meny of technical terms are not clearly/correctly used, i.e. the technical language is not very understandable or makes a chaotic impression.
    • Some illustrating graphs, for example Figure 3 (needs clearly description) and 8, must be improved in terms of the content or/and graphical quality.
    • References to presented Figures/Illustrations/Tables should be consequently included in the text, for example a reference to Figure 1 is missing.
    • Because of the mentioned shortcomings, especially 1.3, it is difficult to evaluate the study accuracy and politeness, as well as the developed tool utility potential.
  • Abstract: Conclusion should be shortly referred to in the abstract.
  • Introduction should be extended and the main intention of the work/study/article should be clear presented/highlighted in here, in the context of some of the Numerous tools have been developed and employed 26 in the last years …” (25-26) which should be shortly presented and referred to or it should be mentioned that they are presented in the following section.

Thus, the structure of Introduction should be thoughtful. It is not very clearly composed, especially 18-92. Section 1.3 Aim and structure of the paper, is not properly located, i.e. it should be located earlier before a review which is a part of the article.

  • Methodology: Main text, illustrations such as for example Table 2, Figures mentioned before should be revised and clearly composed (content, graphics, writing rules, technical terms accuracy and intelligibility…)
  • Results: The title of Table 3 takes more place then the table. (?) It should be replaced with an introduction/description …

Results are not clearly presented.

The comparison between Calliope and MESS could probably be more clear if the compared systems were presented side by side.

  • Discussion/Conclusions: Much attention is paid to the comparison of Calliope and MESS as if it were the purpose of the analysis while possible advantages are not highlighted good enough. MESS is reduced “to the shadow”.

Please, take a look at the title of your article and pay more attention to MESS.

Summary: I firmly believe that the mentioned major improvements are necessary to make the article more attractive for readers and should be provided by the authors before publishing of the article can be re-evaluated.

Reviewer 3 Report

Interesting article

Some comments and questions follow here:

1. Section 1: are you sure there is a lack of simulating models. you found 40 open source models. what is actually meant by mutli energy systems here? are you trying to see each sector separately? are not the different sectors actually interconnected with each others? also optimization models can be actually used for simulationg purposes. how do you consider the socio-technical aspect of the energy system based on the deffinition you give? 

line 112 is unclear. what do you mean by that?

line 123-131: it is still difficult to see the gap this contribution to fill in?

2. Methodology: this section as it is now, is very vague and represents a kind of descriptional work.  Visualization of the methodology through schematic representation which shows the different steps will help a lot to follow this section and the outcome.

With increasing attention given to LCA, how will this model consider such approach?

Contribution: What is actually the current state of the art in this area and in what way one can see the contribution from your study (what is the takeaway from this study?)  Is open source the main contribution? Here it is important that you explain the core contribution of your paper more clearly by comparing with other relevant studies and highlighting distinguishing emphasis and motivation of their approach.

Result: comparing only with one model may not be enough for validation. have you tried the comparison with more models and prsent breifely the outcome? what do you think the outcome will be if compared with existing none open source models?

Accuracy: elaborate also the accuracy of the model

Round 2

Reviewer 2 Report

Thank you for your letter. I think the research and this article intension is presented much more clear for readers now. I am satisfied with the answers as well as the improvments following. Have no more comments, and I think this article is well componed and can be published.

Reviewer 3 Report

There is  a significant improvemnet now.

Will you please update your article by adding mathematical expression of each component in Figure 4.
